# Optimal Halbach Configuration for Flow-through Immunomagnetic CTC Enrichment

**DOI:** 10.3390/diagnostics11061020

**Published:** 2021-06-02

**Authors:** Michiel Stevens, Peng Liu, Tom Niessink, Anouk Mentink, Leon Abelmann, Leon Terstappen

**Affiliations:** 1Department of Medical Cell BioPhysics, University of Twente, 7522 NB Enschede, The Netherlands; p.liu-2@utwente.nl (P.L.); t.niessink@student.utwente.nl (T.N.); a.mentink@utwente.nl (A.M.); L.w.m.m.terstappen@utwente.nl (L.T.); 2Department of Molecular Nanofabrication, University of Twente, 7522 NB Enschede, The Netherlands; 3KIST Europe Forschungsgesellschaft mbH, 66123 Saarbrücken, Germany; l.abelmann@utwente.nl; 4MESA+ Institute for Nanotechnology, University of Twente, 7522 NB Enschede, The Netherlands

**Keywords:** finite element model, magnetic, enrichment, circulating tumor cell, Halbach, flow

## Abstract

Due to the low frequency of circulating tumor cells (CTC), the standard CellSearch method of enumeration and isolation using a single tube of blood is insufficient to measure treatment effects consistently, or to steer personalized therapy. Using diagnostic leukapheresis this sample size can be increased; however, this also calls for a suitable new method to process larger sample inputs. In order to achieve this, we have optimized the immunomagnetic enrichment process using a flow-through magnetophoretic system. An overview of the major forces involved in magnetophoretic separation is provided and the model used for optimizing the magnetic configuration in flow through immunomagnetic enrichment is presented. The optimal Halbach array element size was calculated and both optimal and non-optimal arrays were built and tested using anti-EpCAM ferrofluid in combination with cell lines of varying EpCAM antigen expression. Experimentally measured distributions of the magnetic moment of the cell lines used for comparison were combined with predicted recoveries and fit to the experimental data. Resulting predictions agree with measured data within measurement uncertainty. The presented method can be used not only to optimize magnetophoretic separation using a variety of flow configurations but could also be adapted to optimize other (static) magnetic separation techniques.

## 1. Introduction

The most commonly used technique to isolate circulating tumor cells (CTC) is positive immunomagnetic enrichment. This methodology is simple and effective: Magnetic particles bound to a marker that is present on tumor cells but not on other cells in the suspension are used to label and magnetically separate labeled from unlabeled cell populations.

Due to the low frequency of CTC, the number of patients from which they can be obtained is limited. In recent years, diagnostic leukapheresis (DLA) has been introduced to overcome this limitation of sampling volume z [1,2]. Although small aliquots of DLA product can be successfully processed on the CellSearch system [3,4,5], it is not optimized for samples containing high concentrations of leukocytes without the abundant presence of red blood cells. As a result, aliquots of only 2 × 10^8^ leukocytes, representing ~5% of a DLA product, are used in the standard workflow. To create a more flexible separation system that is not limited by a fixed volume or concentration, several attempts have been made to separate cells from a flow using various flow and magnetic configurations [6,7,8,9,10,11]. This type of separation is useful not only for the enrichment of CTC from blood or DLA products, but also paves the way for systems designed to enrich CTC from blood in an in vivo setting [12].

The success of any magnetic separation is dependent on the sensitivity and specificity with which the magnetic particles bind to the tumor cells, in combination with the ability of the magnetic configuration to attract these particles. Whereas the former depends on the particle characteristics as well as the amount of surface antigens present on the cells, the latter is determined by the magnetic characteristics of the particles and the magnetic configuration used. There are several reviews of magnetic particles [13,14] and descriptions of their performance in different magnetophoretic (flow) setups [6,7,8,9,10]. In order to generate a larger force at a distance in some cases a Halbach configuration is used for static [15] or flow-through separation [11,16,17]. In most of these a general formulation for magnetic force is reviewed, and in some an estimation of the force exerted by the used system is calculated. However, even though the magnetic configuration determines the force exerted on a cell with a given labeling, an optimization of the magnetic configuration is not described.

In this article, we will examine which aspects to consider when optimizing magnet configurations for maximal capture efficiency. We describe the equations involved in the enrichment of immunomagnetically bound tumor cells and model a magnetophoretic flow separation system using COMSOL Multiphysics 5.5 (COSMOL, Stockholm, Sweden). Using this model, we optimize the magnetic configuration to maximize cell separation efficiency. We compare our model results to an independently developed analytical solution programmed in Python and use the COMSOL model results in combination with experimental cell distribution and recovery data to show the validity of this model approach.

## 2. Materials and Methods

### 2.1. Modeled System

The magnetic configurations and flow conditions were modeled and optimized for immunomagnetic enrichment of CTC. This was done for an 800-µm-high, 5-mm-wide and 50-mm-long flow channel with a 200-µm-thick wall and a flow rate of 1 mL/min. An array of magnets is placed against this channel extending out beyond the channel length. Under the assumption that the flow profile and magnetic forces are constant over the width of the channel, we used a 2D model in our approach. Figure 1 shows a schematic drawing of the magnetophoretic principle and modeled setup.

#### Forces

The trajectory of all cells in a magnetophoretic flow separation system are the result of a combination of several forces, the most important of which are the magnetic-, drag- and gravitational forces. Additional forces are hydrodynamic lift, electrostatic, Van der Waals and Brownian motion [18]. As these latter forces are several orders of magnitude lower, we take only the magnetic, drag and gravitational forces into account and assume the other forces to be negligible.

### 2.2. Magnetic Force

The magnetic force exerted onto a cell is dependent on the number of particles bound to the cell, the size, material and crystalline structure of the individual particles, as well as the magnetic flux and flux gradient resulting from the magnetic configuration used. In general, the magnetic force $F_{m}$ (N) on a small particle with uniform magnetization M (A/m), volume $V_{p}\;$(m^3^) and moment $m=MV_{p}$ (Am^2^) in a magnetic field ***B*** (T) is given by:(1)$$F_{m}=\left(m\cdot \nabla \right)B$$

Equation (1) clearly indicates that in order to maximize the magnetic force, both the magnetic moment and the magnetic field gradient are important.

### 2.3. Magnetic Moment

Under influence of the magnetic field, the magnetic domains in the magnetic nanoparticles (MNP) will align themselves with the external field, causing the particles to become magnetic. When the external magnetic field strength $H_{ext}$ (A/m) increases, the magnetization M (A/m) also increases as more and more domains within the particle become aligned. If the external magnetic field strength is increased further, there will be a certain point at which all domains will align with the external field and the magnetization will plateau. This maximal level of magnetization is known as the saturation magnetization $M_{s}$ (A/m) and is dependent on the material and crystalline structure of the MNP. This is graphically depicted in Figure 2.

Often the equations used to describe the magnetic forces involved in immunomagnetic enrichment are either based on the assumption that the magnetic moment of the particles is fully saturated [6,16,19], or a constant value for the particles magnetic susceptibility is assumed [7,9,10,11,15,20]. Articles in which both domains are acknowledged are for instance those by Furlani [21], Joshi [22], Hoyos [16] and Shevkoplyas [23]. The latter also takes into account the remnant magnetization arising from their use of not fully superparamagnetic particles and shows good agreement with experimental data.

The relation between magnetic field and magnetization for CellSearch ferrofluid, along with the approximations used later on in this paper are shown in Figure 3.

### 2.4. Magnetic Gradient

The magnetic force is determined by the magnetization of the particle in combination with the magnetic field gradient. The gradient is determined by the change in magnetic field strength over distance, which is why a pair of magnets with opposing orientations is used in many magnetic separation systems. In particular in proximity to where the alternatingly oriented poles meet, the magnetic field strength changes rapidly, resulting in a high gradient. Generally, the more of these changes in orientation are realized on a surface, the greater the force will be close to the surface. However, as a large gradient is the result of a fast decrease in magnetic field strength, it also means that the force does not extend very far from the surface.

The magnetic flux and flux gradient as a result of 4-mm, 2-mm and 1-mm-wide magnets were calculated using COMSOL and are shown in Figure 4. It can be seen that a decrease in magnet size leads to an increase in the magnetic gradient ∇B in the region close to the magnets. This large magnetic gradient is attained at the cost of a lower magnetic field B in the top portion of the shown channel. By placing a magnet with a perpendicular orientation between the alternating magnets, we force the magnetic field lines out of the array more prominently on one side. This causes an increase of magnetic field strength and gradient on one side of the array. This configuration is known as a Halbach array, and the resulting field (gradient) for a 2-mm Halbach is shown in the right-hand panel of Figure 4.

### 2.5. Magnetic Force Calculation

In order to calculate the magnetization of a particle and its resulting magnetic force we make the following simplifications:The particle is a perfect sphere with a uniform magnetization parallel to the external field.The surrounding liquid is non-magnetic.There are no unbound particles present.

Using the first simplification of a parallel magnetization, one can show that Equation (1) simplifies to:(2)$$F_{m}=m\;\nabla \left(B\right)=\frac{1}{2}V_{p}\frac{M}{B}\;\nabla \left(B^{2}\right)$$
where we use the vector identity $B\cdot \nabla \left(B\right)=\frac{1}{2}\;\nabla \left(B^{2}\right)$.

The relationship between the magnetization of a spherical particle and the applied field is given by Equation (3), where the dimensionless variable $\chi_{p}$ denotes the particles’ magnetic susceptibility and $\mu_{0}$ (=1.26 µN/A^2^) the vacuum permeability.
(3)$$M=\frac{3\chi_{p}}{\mu_{0}\left(3+\chi_{p}\right)}B$$

Even though not used here, it is worth noting that, in biologically oriented papers, Equation (4) is used to describe the magnetic force. It is based on a simplification by stating that, when considering the cell and attached magnetic particles as a single magnetic entity, the average $\chi_{p}$ becomes very small ($\chi_{p}\ll 1)$. Using this reasoning one can state that 3 +$\;\chi_{p}$ ≈ 3, leading to:(4)$$F_{m}=\frac{1}{2}V_{p}\frac{\chi_{p}}{\mu_{0}}\;\nabla \left(B^{2}\right)$$

This simplification is not needed here. By introducing the relative permeability of the particle $\mu_{p}=\chi_{p}+1$ and combining Equations (2) and (3) we obtain:(5)$$F_{m}=\frac{1}{2}V_{p}\frac{3\left(\mu_{p}-1\right)}{\mu_{0}(\mu_{p}+2)}\nabla \left(B^{2}\right)$$
for the magnetic force on a particle. This is under the assumption of a non-magnetic fluid ($\mu_{f}=1)$ and using the particle radius *R* (m) and magnetic field intensity $H=\frac{B}{\mu_{0}}$ (A/m) equal to the expression COMSOL uses to calculate the magnetic force:(6)$$F_{m}=2\pi R^{3}\mu_{0}\mu_{f}\frac{\mu_{p}-\mu_{f}}{\mu_{p}+2\mu_{f}}\nabla \left(H^{2}\right)$$

The magnetic field and its gradient are calculated based on the magnetization of the magnets, leaving the last unknown to be the magnetization of the particles.

The cell and bound particles are modeled as a single magnetic entity which is subjected to the described forces. Due to COMSOL using the same particle radius for the magnetic and drag force, we will have to use an effective magnetization that in combination with the cell radius results in a total magnetic moment equal to that of the particles. As the magnetization is dependent on the magnetic field *B*, we introduce an approximation function for the effective magnetization of the cell using an arctangent function:(7)$$M_{\mathrm{eff}}=\frac{2}{\pi }M_{s,\mathrm{eff}}\tan^{-1}\left(\frac{B}{B_{s}}\right)$$
where $M_{s,\mathrm{eff}}$ (A/m) is the effective saturation magnetization of the cell and $B_{s}=22\;\mathrm{mT}$ is a fitting parameter based on the measured magnetization curve of CellSearch ferrofluid (Figure 2).

Using the approximation function (7) we can express the effective permeability of the cell as a function of the *B*-field in order to be able to implement this in COMSOL.
(8)$$\mu_{p,\mathrm{eff}}=\frac{3B+2\mu_{0}M_{\mathrm{eff}}\left(B\right)}{3B-\mu_{0}M_{\mathrm{eff}}\left(B\right)}$$

### 2.6. Drag Force

The other major force involved is the drag force, which is a result of the fluid flow around the particle. As high magnetic gradients are known to extend only a very limited distance from the magnetic material a shallow flow channel is used in the form of an Ibidi µ-channel slide (Ibidi, Gräfelfing, Germany). These channels are 50 mm in length, 5 mm in width and available with channel heights of 200, 400, 600 or 800 µm. For this optimization we modeled the 800-µm channel. Together with their commercial availability, the advantage of these flow slides in this context is their thin (0.18 mm) bottom, which allows the magnetic configuration to be placed in close proximity to the fluid flow.

As these channels are quite shallow, we expect a laminar flow pattern. The drag force on a spherical object such as a cell in a laminar liquid is described by Stokes’ law using the viscosity η (Pa∙s), particle radius *r* (m) and velocity of the particle $v_{p}$ and fluid $v_{f}$ (m/s) as follows:(9)$$F_{d}=6\pi \eta r(v_{p}-v_{f})\;$$

To estimate the validity of this assumption of laminar flow, we calculate the Reynolds number as follows:(10)$$Re=\frac{QD}{\eta A}\rho$$
where *Q* is the fluid flow (m^3^/s), *A* is the cross-sectional area of the channel (m^2^) and ρ the fluid density (kg/m^3^). The characteristic length *D* is for a rectangular channel given by $\frac{2*height*width}{height+width}$.

For the 800-µm channels, using *Q* = 1 mL/min, ρ = 1000 kg/m^3^ and η = 0.89 mPa∙s results in a Reynolds number of 6.5, well within the laminar flow domain.

### 2.7. Gravitational Force

The gravitational force is a result of the difference in density between the cell and the surrounding media. This difference causes the gravitational force exerted onto the cell to differ from the buoyancy force exerted by the surrounding liquid. In the literature, these forces are sometimes described separately or taken together as either the gravitational or buoyancy force. Here we use the following definition:(11)$$F_{g}=g*V_{p}\left(\rho_{c}-\rho_{f}\right)$$
where g = 9.81 m/s^2^ and, $\rho_{c}=1070\;\mathrm{kg}/m^{3}$ and $\rho_{f}=1000\;\mathrm{kg}/m^{3}$ are the density of the particle and fluid.

### 2.8. COMSOL Model

For the COMSOL simulations we used the Laminar Flow interface of the Computational Fluid Dynamics module to simulate the fluid flow. The material was set to water as defined by the COMSOL default. For the flow inlet we used a mass flow rate of 1 g/min while the outlet was set to prevent back-flow. The boundary condition on the channel walls was set to no slip.

The magnetic fields were computed using the AC/DC modules’ “Magnetic fields, No currents” interface. The model is enclosed by a magnetic insulation boundary at 45 mm distance in the vertical direction and 85 times the simulated magnet width in the horizontal direction. The flow-channel and surrounding air were set to a relative permeability of 1.

For the simulation of particle trajectories, we used the Fluid Flow interface of the Particle Tracing module. The particle distribution as well as the initial particle velocity was set to reflect the fluid flow profile. An accumulator was used to count the number of particles reaching the channel surface. Wall conditions were set to freeze.

Meshing was done using a maximal mesh element size for the boundary mesh on the flow channel walls of 0.05 mm together with corner refinement and a boundary layer for no slip walls. Remaining geometries were meshed using element size parameters calibrated for finer general physics.

### 2.9. Model Validation

Even though the COMSOL program is easy to use, it also has the capability to generate results based on unintentional assumptions or presumed conditions. We checked the COMSOL model with an analytical model in which we assumed that the particles are always saturated, and that the flow profile in the channel is parabolic. The magnetic field gradient is calculated by integrating over the charge densities. Particle trajectories were calculated using the solve_ivp routine of the Python scipy.integrate package. The COMSOL results were accurately reproduced, with differences smaller than 2% around the 50% recovery points (See Table A1 in Appendix B). The COMSOL model is available as Appendix A, the Python source code can be downloaded from GitHub (https://github.com/LeonAbelmann/Trajectory.git).

### 2.10. Experimental Validation

To validate the approach experimentally, we tested the following configurations.

The optimized Halbach array with 12-mm-long, 1-mm-wide, N52 magnets with a height of 2 mm (horizonal magnetization) and 2.75 mm (vertical magnetization) (Risheng Magnets, Ningbo, China).A Halbach array consisting of three rows of commercially available 5 × 1 × 1.5 mm N45 stock magnets, (Supermagnete, Gottmadingen, Germany).A Halbach array consisting of commercially available 15 × 4 × 4 mm N45 stock magnets, (Supermagnete, Gottmadingen, Germany).

All arrays were assembled on soft magnetic sheets after which a 3D-printed plastic support was glued to the backside of the array. The soft magnetic material was subsequently removed, allowing the flow channel to be placed directly against the magnet surface. The three magnetic arrays used in our experiments are shown in Figure 5.

To test the different configurations, we used cells obtained from three different prostate cancer cell lines: PC3, PC3-9 and LNCaP. Cells were cultured in RPMI1640 (Lonza, Basel, Switzerland) supplemented with 10% FBS (Sigma-Aldrich, St. Louis, MO, USA) and 1% penicillin/streptomycin (Lonza, Basel, Switzerland). Upon reaching 70–80% confluence they were trypsinized using 0.05% trypsin-EDTA (Gibco, Waltham, MA, USA) and fixated using 1% formaldehyde.

The recovery of each cell type is expected to be dependent on the number of magnetic particles bound to the cells. This level of particle binding is determined in turn by the amount of target antigens present on the cell surface. For this reason, we first measured the level of EpCAM expression of these cell lines by staining the cells with anti-EpCAM(Vu1d9)-PE (Sigma-Aldrich, St. Louis, MO, USA). The PE intensity was measured and quantified by flowcytometry (BD FACS Aria II) using BD Quantibrite™ Beads PE Fluorescence Quantitation Kit (BD, Franklin Lakes, NJ, USA).

Additionally, we measured the (relative) ferrofluid labeling of these three cell lines. To do so we labeled the cells by incubating them with CellSearch ferrofluid (Menarini, Bologna, Italy). We subsequently centrifuged and washed the cells to remove unbound ferrofluid. The bound ferrofluid was then stained using anti-mouse IgG PE (Sigma-Aldrich, St. Louis, MO, USA), which binds to the mouse-based anti-EpCAM on the ferrofluid surface. The fluorescence intensity arising from the ferrofluid labeling was measured using flow cytometry. To evaluate recovery, the cell lines were pre-stained using either CellTracker Orange, CellTracker Green or CellTracker Deep-Red (Thermo Fisher Scientific, Waltham, MA, USA).

Approximately 40,000 cells of each type were incubated with CellSearch ferrofluid (Menarini, Bologna, Italy), at a concentration of 15 µL of ferrofluid per ml of sample for three times ten minutes in a BD iMag Cell Separation magnet (BD, Franklin Lakes, NJ, USA). Samples were mixed in-between. Cells were subsequently centrifuged to remove unbound ferrofluid particles and resuspended in casein buffer. We split the sample into four portions. For each of the three configurations one portion was flowed through an 800-µm high Ibidi flow channel positioned directly against the magnetic array, at a flow speed of 1 mL/min. The fourth portion was used to determine the concentration. After the samples passed, the channel was rinsed using 2 mL of PBS at 2 mL/min. The magnetic array was removed and the enriched fraction was flushed out into a TruCount tube (BD, Franklin Lakes, NJ, USA). The recovery of each cell population was counted using flow cytometry (BD FACS Aria II).

## 3. Results

### 3.1. Magnet Width Optimization

Simulated cells were defined as spheres with a diameter of 10 µm and a density of 1077 kg/m^3^. The magnets were modeled as 1.5-mm-high N52 grade permanent magnets, and the magnetophoretic force was simulated based on a presumed total magnetic moment of 10 fAm^2^ per cell. The cells were distributed over the inlet based on the simulated flow profile. Recovery was determined as the percentage of cells reaching the surface of the channel at which the magnet array is positioned. To examine the difference between our approximation and the assumption of a saturated magnetic moment or fixed permeability we determined the recovery as a function of the magnet width for all three options at 0.05 mm intervals in the range of 0.4 to 2.0 mm (Figure 6). It can be seen that the assumption of a saturated magnetic moment seems to be a usable simplification when optimizing a magnetic configuration because the found optima are almost identical.

As cells starting at the very top of the channel are not captured when a configuration with a maximal recovery of <100% is modeled, the optimum size will also be dependent on the height of the channel from which a cell can be captured. For instance, when a magnetic moment is modeled that results in a maximal recovery of 50%, the optimization will result in a configuration that is optimal for only this portion of the channel. For this reason, one must optimize using a modeled magnetization which results in a maximal recovery close to 100%.

To test whether a relation similar to the often used rule that the size of magnet should be the same as the distance at which a force is to be generated can be used when a Halbach array element width, we calculated the optimal magnet width for different channel heights. Figure 7 shows that within the calculated range, there is a linear relation between the optimal magnet width and the flow channel height when using 1.5-mm-high magnets and a 200-µm wall thickness.

### 3.2. Magnet Height

The magnetic force increases with magnet height as more magnetic material is added. However, as the additional magnetic material is at a greater distance its benefit quickly diminishes for both the vertically and horizontally oriented magnets. We calculated the recovery for magnets heights of 0.5 to 3.0 mm at 0.25 mm intervals to see the influence of the magnet height (Appendix C, Table A2). To have sufficient magnet surface available for strong adherence to the substrate we chose to use different magnet heights. Considering that the array becomes more difficult to assemble when using larger magnet heights we chose to construct the optimized array using vertically oriented magnets with a height of 2.75 mm and horizontally oriented magnets with a height of 2 mm.

As magnet height can have an influence on the optimal width of the array we simulated the recovery for different magnet widths using the chosen magnet heights. This increase in magnet height resulted in a shift of the optimal width from 0.9 mm to 1.0 mm (Appendix D, Figure A1).

### 3.3. Halbach versus Converntional Alternating Array

To check that the Halbach array is an improvement over the conventional alternating array, we also performed the optimization for alternating arrays using 2.375-mm-high magnets, thereby using the same magnet volume as used in the optimized Halbach array. This resulted in an optimal magnet width of 2.0 mm (Appendix D, Figure A1). As the magnetic moment of the cells is unknown, we calculated recoveries as a function of magnetic moment. The recovery calculated for the optimized alternating array shows a minimal magnetic moment needed for 100% recovery of 15 fAm^2^ compared to 11 fAm^2^ for the optimized Halbach array, see Figure 8. The higher gradient of the alternating array close to the surface does results in a slightly higher capture for very low magnetic moments, when only the cells in the bottom part of the channel can be captured.

### 3.4. Halbach Array Comparison

We calculated recoveries as a function of magnetic moment for the three different magnetic configurations (see Figure 9). Here the optimized Halbach array is predicted to increase recovery for cells with a magnetic moment up to 15 fAm^2^ when compared to the 1 × 1.5-mm array and up to 24 fAm^2^ when compared to the 4 × 4-mm Halbach array. At higher magnetic moments all arrays are predicted to capture 100% of the cells.

### 3.5. Experimental Validation

The average intensity of EpCAM staining for PC3, PC3-9 and LNCaP was determined to be 7100, 19,700 and 628,400, respectively. The recoveries obtained using the different magnetic configurations are shown in Figure 10. For the PC3 and PC3-9 cells it is clear that there is an increase in recovery when moving towards the optimal array, in agreement with calculations. For the LNCaP cells the recovery is already at 90% for the 4 × 4 mm array, leaving little room for improvement.

As the magnetic moment is not the same for each cell in the population, the total capture efficiency must be calculated using the distribution of magnetic moments of the cells together with the modeled capture efficiency as a function of that moment (Figure 9). We obtained an estimate of the distribution of magnetic moment by fluorescently labeling the bound particles. The resulting histograms showing the fluorescence intensity distributions for PC3 and PC3-9 cells together with the corresponding controls (stained using anti-mouse IgG-PE but containing no ferrofluid particles) are shown in Figure 11A,B. The mean intensity for LNCaP was more than 30 times higher than that of PC3-9, making it likely that its magnetic moment is high enough to ensure efficient capture using all arrays, which agrees with the measured recovery. We therefore only considered PC3 and PC3-9.

We assume that fluorescence intensity scales linearly with the magnetic moment. We therefore multiplied each element in the scaled distributions of PC3 and PC3-9 with the corresponding calculated recovery (Figure 11C) and fitted the values to the experimental data shown in Figure 10. We obtain the best fit when using a scaling factor of 120 units of fluorescence intensity per fAm^2^ and an offset of −12 and −6.5 fAm^2^ for PC3 and PC3-9, respectively. The need for an offset is expected to be a result of the non-specific binding of anti-mouse-IgG-PE to the cells, causing a background intensity. This agrees with the observation that the offsets to obtain the best fit are approximately equal to the peak intensities of the negative control samples. To calculate cell recoveries the obtained negative magnetic moments are set to zero.

The resulting calculated recoveries, as shown in Figure 11D, agree within measurement uncertainty with the observed values of both cell lines and all three arrays.

## 4. Discussion

Leukapheresis is frequently used to harvest hematopoietic stem cells from peripheral blood for stem cell transplantation. These hematopoietic stem cells are contained within the cell population expressing the CD34 antigen present in < 1% of the leukocytes and can be harvested by flow-through immunomagnetic cell separation for peripheral blood stem cell separation [24]. In leukapheresis products collected for autologous stem cell transplantation, circulating tumor cells have been observed, raising the concern that they might be seeds for metastasis [25,26,27]. On the one hand, these observations have led to efforts to further purify the hematopoetic stem cells [28]. On the other hand, to the development of technologies to detect rare cancer cells in blood and leukapheresis products for diagnostic purposes [29,30,31,32]. For diagnostic purposes the percentage of patients in which ten or more CTC can be detected in a tube of blood for real-time characterization of cancer is too low [1], leukapheresis provides the opportunity to overcome this limitation [2]. Flow-through immunomagnetic separation is an attractive means to process not only blood but also the larger volumes obtained through leukapheresis, and several approaches for flow-through magnetic enrichment have been introduced [8,9,33,34,35]. We opted to develop a COMSOL model to guide us to design, build and test flow through immunomagnetic separation of cancer cells using commercially available flow channels, an external magnetic array and magnetic particles directed against the EpCAM antigen.

The COMSOL model is simplified because no unbound ferrofluid particles are assumed to be present, even though the vast majority of particles is unbound in a regular magnetic separation. These particles will attract each other and thereby increase the number of particles attached to the cell during the separation. We assume that this effect takes place as soon as the particles are in reach of the field and that the effective number of particles bound to the cell will include these magnetically loaded particles.

The results show that by improving the magnetic Halbach array recovery is increased for the PC3 and PC3-9 cell lines expressing low levels of EpCAM antigens, whereas there is little benefit for the recovery of LNCaP cells with relatively high levels of antigen. However, as the EpCAM expression of patient CTC has been shown to be similar to that of PC3-9 cells [36], this is also the range of interest when evaluating an EpCAM-based CTC enrichment.

The resulting scaled distribution of magnetic moments of PC3 and PC3-9 cells that best fits the measured recovery indicates that there is substantial population with no or very little magnetic particles attached. This would mean that for these populations a further optimization of magnetic separation must focus on the increase of magnetic labeling, because 24% of the PC3 cells are modeled to have a negative magnetic moment. For instance, an increase could be obtained by using the controlled aggregation of ferrofluid that is normally used in the CellSearch system [37]. Ideally, one would know the distribution of magnetic moment for the patient CTC, allowing the array to be optimized using this information. We show an increase in recovery as a result of the described optimization using cells in buffer, whereas the goal of this setup is to enrich rare CTC from highly concentrated DLA samples. Due to the large background of unlabeled cells, it will be more difficult for the cell to reach the wall of the channel in these samples. Even though this effect will probably reduce the overall recovery, this optimization shown here is expected to have a similar impact on the recovery in these samples.

Using the optimized configuration described it is possible to process an entire DLA sample because the flow-based configuration does not impose a limit on sample volume. Once all sample has been flowed through the channel, the magnet array is removed and captured cells simply washed out. In case of these large samples, which will also contain unbound ferrofluid, accumulation of ferrofluid will occur. As the separation progresses it is likely that this accumulated ferrofluid will have a shielding effect on the magnetic field, thereby decreasing the force exerted as more sample is processed. In this case it will be necessary to rinse the collected cells and ferrofluid out of the channel at regular intervals, thereby allowing the processing of a complete DLA sample.

## 5. Conclusions

We used a finite element model to calculate the recovery of cells with bound magnetic particles in a flow channel. The magnetic force was calculated using an approximation of the measured magnetization curve of the particles. We show that a high field approximation assuming magnetic saturation predicts optimization results within 5% of the magnetization curve model. However, there is no noticeable improvement in computation time as a result of the simplification, making it useful only in cases where the magnetization curve is unknown. A low field approximation assuming a linear permeability however results in a perceived optimum at a ~30% larger element size and should be avoided.

When using a Halbach array, there is an optimum in capture efficiency with respect to the dimensions of the magnets. The optimal magnet width increases linearly with increasing channel height. For an 800-µm-high, 5-mm-wide channel with a flow of 1 mL/min our model predicts a minimum magnetic moment of 15 fAm2 needed to obtain 100% recovery when using an array made of 1.0 × 1.5 mm stock magnets. By optimizing the magnet width and using custom 1-mm-wide and 2/2.75-mm-high elements the minimum magnetic moment needed for 100% recovery is lowered to 11 fAm^2^.

We compared our model predictions to experimental results using three cell lines with different EpCAM expressions. As predicted, the optimization increased recovery when cells with a low EpCAM expression are used. We show that the distribution of magnetic moments between cells must be considered. We obtain a distribution of the magnetic moments of the cells by scaling the fluorescence intensity of labeled cells and show that the model predicts capture efficiencies within measurement uncertainty.

By optimizing the capture efficiency of a flow-based enrichment system capable of processing a complete DLA sample, we are one step closer to unlocking the potential of DLA as a means to capture sufficient CTC for therapy monitoring, guidance and tumor characterization in all cancer patients.

## Figures and Tables

**Figure 1 diagnostics-11-01020-f001:**
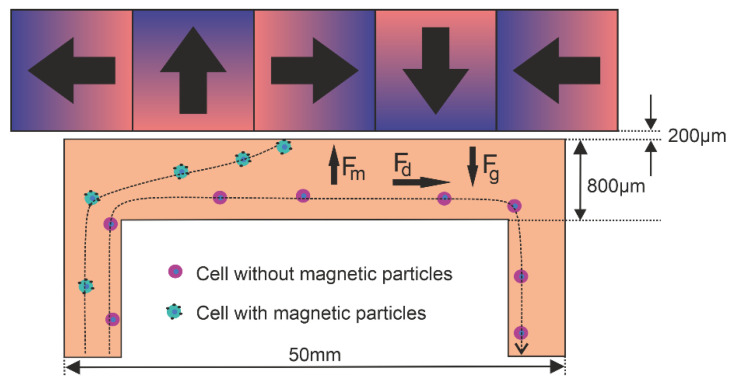
Schematic representation of the magnetophoretic flow separation setup comprised of a flow channel with a Halbach array showing the magnetic (***F***_m_), drag (***F***_d_) and gravitational force (***F***_g_).

**Figure 2 diagnostics-11-01020-f002:**
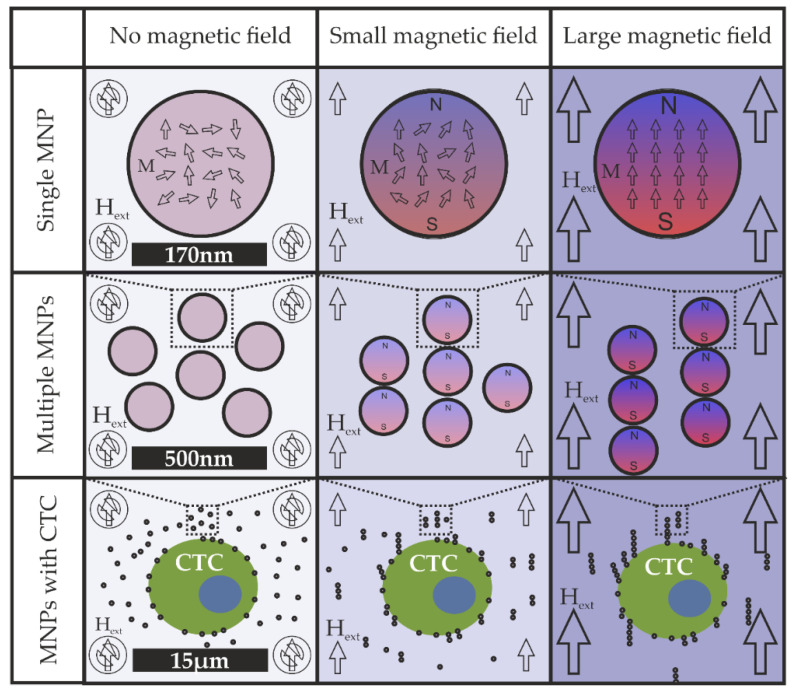
Graphical depiction of the magnetization of a magnetic nanoparticle (MNP) in an increasing external field. Top panel, alignment of magnetic domains in the magnetic particles. Middle panel, alignment of magnetic particles. Bottom panel, alignment of magnetic particles in the presence of a CTC.

**Figure 3 diagnostics-11-01020-f003:**
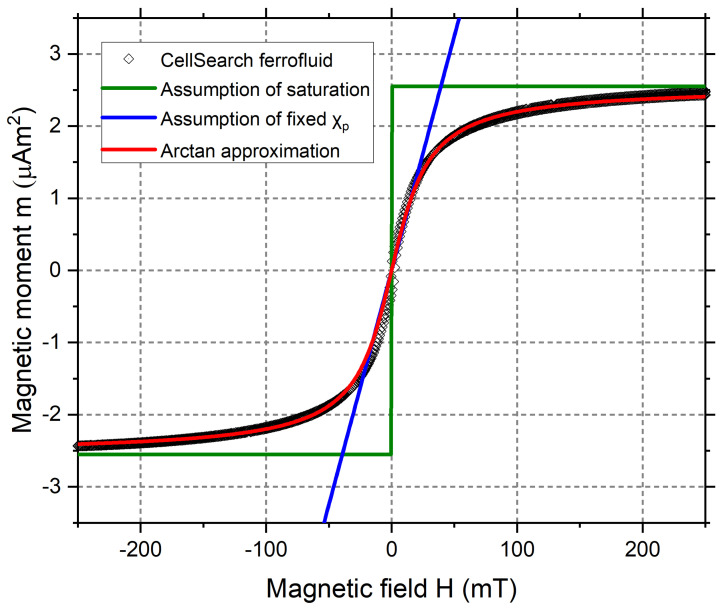
Measured magnetization curve of CellSearch ferrofluid and the approximations used.

**Figure 4 diagnostics-11-01020-f004:**
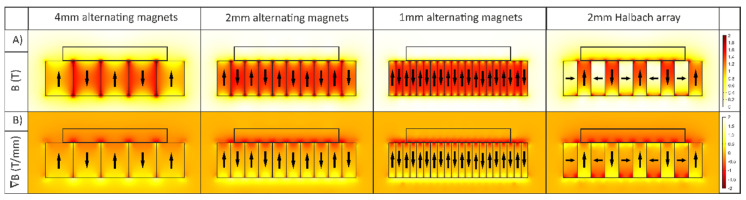
COMSOL simulation of (**A**) the magnetic field and, (**B**) the field gradient, resulting from magnets in an alternating orientation and Halbach array.

**Figure 5 diagnostics-11-01020-f005:**
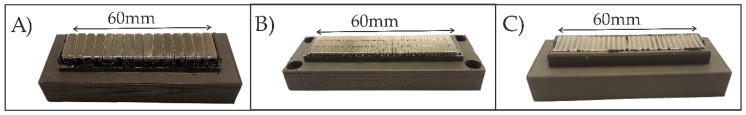
Halbach array assemblies made using (**A**) 15 × 4 × 4 mm; (**B**) 5 × 1 × 1.5 mm; (**C**) 12 × 1 × 2.75 mm/12 × 2 × 1 mm magnets.

**Figure 6 diagnostics-11-01020-f006:**
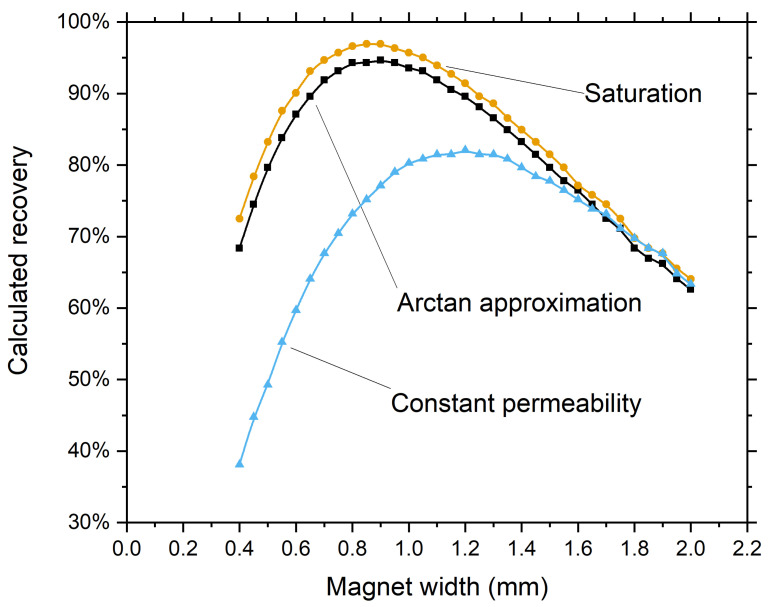
Calculated recovery versus magnet width for a 1.5-mm-high Halbach array using different approximations, showing a similar optimal magnet width for the assumption of saturation and the arctangent function but a clearly deviating optimum when assuming a constant permeability.

**Figure 7 diagnostics-11-01020-f007:**
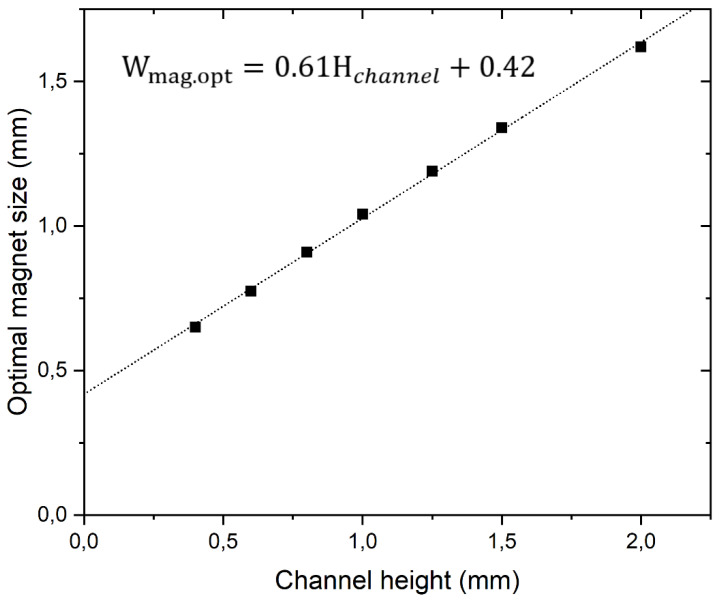
Calculated optimal magnet width for Halbach arrays consisting of 1.5-mm-high elements in flow channels of increasing heights. The optimal width shows a linear relation to the channel height.

**Figure 8 diagnostics-11-01020-f008:**
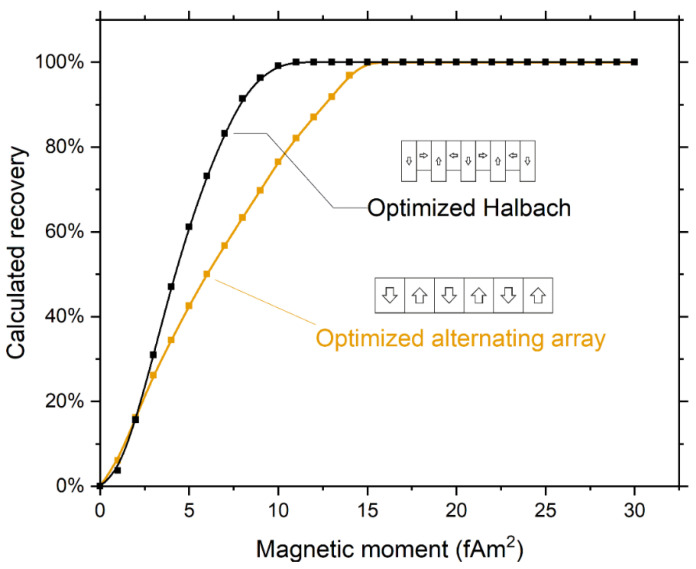
Comparison of recovery dependence on magnetic moment between the optimized alternating array (2.0 mm width) and the optimized Halbach array (1.0 mm width) showing a decrease in magnetic moment needed to achieve 100% recovery. For very low magnetic moments, where only the cells close to the surface are captured, the alternating array outperforms the Halbach array due to its higher gradient close to the surface.

**Figure 9 diagnostics-11-01020-f009:**
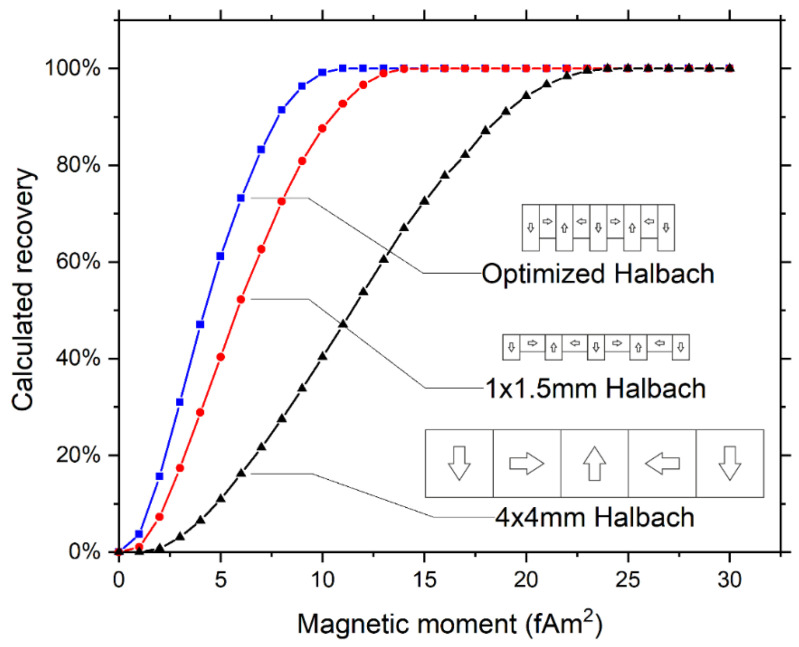
Calculated recoveries using different magnetic arrays showing a decrease in the magnetic moment needed to achieve 100% recovery when magnet elements with a smaller width are used. The optimized Halbach configuration has vertically oriented magnets with a width of 1 mm, a height of 2.75 mm, whereas horizontally oriented magnets have a width of 1 mm and height of 2 mm.

**Figure 10 diagnostics-11-01020-f010:**
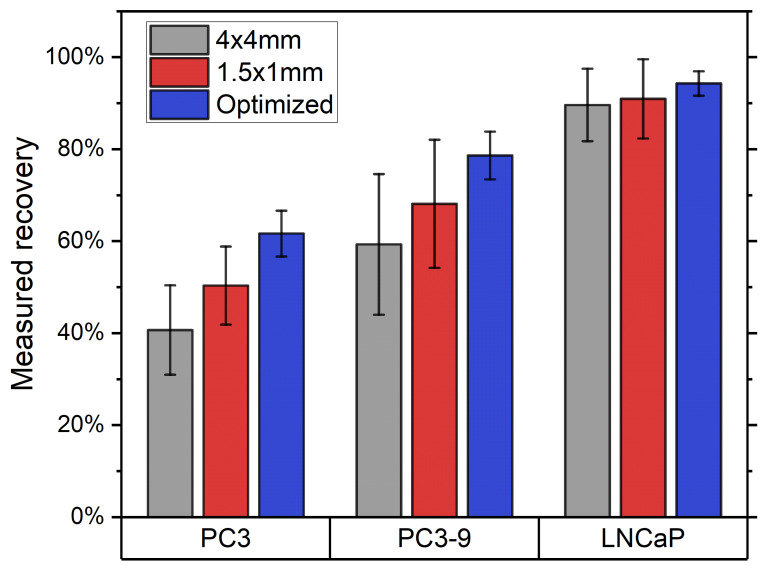
Recoveries (mean and SD) of the three cell lines (N = 4) for all three magnet arrays show a clear increase in recovery for the PC3 and PC3-9 cells when using an improved array. As almost all of the LNCaP cells are already maximally captured using the 4 × 4 mm array there is little further improvement possible.

**Figure 11 diagnostics-11-01020-f011:**
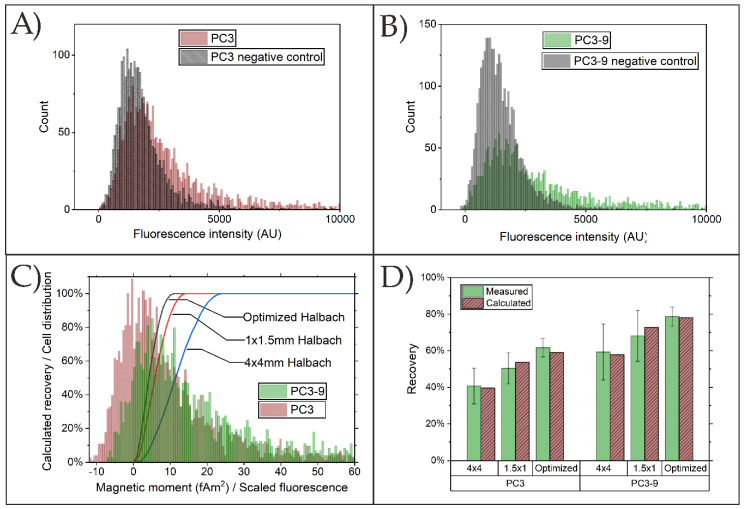
Fluorescence intensity of ferrofluid bound to (**A**) PC3; (**B**); PC3-9, showing clear overlap with negative control. (**C**) Combination of calculated recoveries and scaled ferrofluid distribution at best fit with experimental data. (**D**) Measured recoveries compared to recoveries calculated using the scaled distributions.

## Data Availability

All data is described in the paper or Appendix A.

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
