# Peer review of "Optimal Halbach Configuration for Flow-through Immunomagnetic CTC Enrichment"

_diagnostics, 2021, doi:10.3390/diagnostics11061020_

Round 1

Reviewer 1 Report

It is very important to notice that EpCAM expression levels of tumor cells change according to the gradient changes in the magnetic field force intensity. That is, how many of the magnetic particles (that are surrounded by anti-EpCAM), could bind to the EpCAM on the surface of the tumor cells, depends upon how the expression of the EpCAM is in the tumor cells. Therefore, to illuminate how the EpCAM expression in the tumor cells changes in the different gradient magnetic field force intensities, is a very important task in the near future. If they do so, more accurate optimization will be done in recovering circulating tumor cells from human blood in the upcoming years.     

Author Response

Dear reviewer,

Thank you for your feedback and suggestions.

In light of your suggestion for additional English editing we have employed a professional proofreading service and made numerous improvements.

We agree that the expression of the used surface marker is of great importance for positive magnetic enrichment, indicating the need for a better understanding of the factors that influence their expression in different conditions, including magnetic field force intensities.

Reviewer 2 Report

The work put into this paper is extensive and extremely technical. Application to actual use would require increased volumes of blood if one would want to capture more CTCs in the real life setting. This would not be practical. An alternate capture system between two slides is done  with the Cynvenio CTC system versus the CellSearch system and this closely resembles this system.

I am not clear where this application would be used and the amount of blood proposed to use in the patient setting.

Author Response

Dear reviewer,

            Thank you for your review. We agree there is a resemblance to the no longer available Cynvenio CTC system, that used a similar flow chamber with embedded micromagnetic elements for CTC enrichment from blood.

An increase in blood is clearly needed to capture more CTC in the real life setting. This increase of sampling volume will be achieved through Diagnostic Leukapheresis, which allows the processing of in excess of 5 liters of blood. The proposed amount to use in the patient setting would be in this range, resulting in approximately 100ml of DLA product.

Reviewer 3 Report

The paper describes an optimized Halbach array for a flow immunomagnetic CTC cell seperation method. The research is interesting, Although the manuscript is well-constructed, the are some issues needed to be addressed before, I can recommend this to be published. in summary, The literature review part that using Halbach and/or optimized Halbach array for cell separation is missing, the methodology of Halbach array optimization is not logically stated and clearly explained. The novelty of the method is needed to emphasized in the introduction part too. detail of COMSOL simulation is not described, mesh refinement assay is missing, and the negative control of using a conventional standard array for recovery rate test is missing. 

Below are detailed comments for major issues:

  1. There is no clearly stated methodology and calculation that shows how the optimization of the width, height, and length of the magnet unit is optimized. COMSOL simulation in figure 4 only shows 1 Halbach array design. How the conclusion of optimal size 2mmX2.75mm is made?

  2. Many papers did similar works, either using Halbach and/or optimized Halbach array for cell separations. authors should include them in the literature review part.
    DOI: 10.1016/j.jmmm.2010.11.051
    https://doi.org/10.1051/matecconf/201815306008 https://doi.org/10.1039/C9RA08285A, https://doi.org/10.1021/acs.analchem.5b02431;DOI: 10.1051/matecconf/201815306008; https://aip.scitation.org/doi/10.1063/1.4952612, DOI: 10.1021/acs.analchem.5b02431
     In addition, the authors need to emphasize the novelty and significance of their optimization method in this field. 
  3. Although the COMSOL file is attached in the ESI, The authors still need to describe the parameters boundary conditions as well as the mesh refinement assay in the main manuscript.
  4. The recovery rate of using an optimized Halbach array should also be compared with a conventional standard array method.

Minors:

Line 4, "Peng Lui" please make sure you spell his name correctly.

Line 10,  "low frequency" should be "low presenting frequency" or "low concentration"

Line 292, "40.000" should be "40,000" please correct the rest same problems in context.

Author Response

Dear reviewer,

Thank you for the extensive and very constructive review of our article. We will address the points one-by-one.

  1. We have clarified that there is no optimum in the magnet height, there is only a diminishing effect of its increase. We also added that the heights used are chosen based upon the notion that the added effect of more material would add only little additional recovery while increasing the difficulty of assembly as well as array cost. An overview of the relation between recovery and magnet height was added as an appendix.
  2. We have added textual as well as citation references to reflect the previous work on Halbach arrays for cell separation to the introduction as suggested.
  3. An elaboration on the model including boundary conditions and mesh settings used was added to the main text.
  4. This is a great point. Due to our assumption of an increased benefit of using a Halbach array we neglected to show the difference in a comparison. In our revision we have added the optimization of an alternation orientation array and compare this to the optimized Halbach array using equal magnet volumes.

Thank you for pointing out the spelling mistake in the second authors name.

In order to avoid confusion we have chosen to use the space as a thousands separator in agreement with ISO standards.

Round 2

Reviewer 3 Report

My questions are well answered and I would like to suggest this work publish on Diagnostics